# Testing of the Survivin Suppressant YM155 in a Large Panel of Drug-Resistant Neuroblastoma Cell Lines

**DOI:** 10.3390/cancers12030577

**Published:** 2020-03-02

**Authors:** Martin Michaelis, Yvonne Voges, Florian Rothweiler, Fabian Weipert, Amara Zia-Ahmad, Jaroslav Cinatl, Andreas von Deimling, Frank Westermann, Franz Rödel, Mark N. Wass, Jindrich Cinatl

**Affiliations:** 1Industrial Biotechnology Centre and School of Biosciences, University of Kent, Canterbury CT2 7NJ, UK; M.Michaelis@kent.ac.uk (M.M.); M.N.Wass@kent.ac.uk (M.N.W.); 2Institut für Medizinische Virologie, Goethe-Universität, 60596 Frankfurt am Main, Germany; yvonnevoges@gmx.de (Y.V.); f.rothweiler@kinderkrebsstiftung-frankfurt.de (F.R.); amara_zia12@yahoo.de (A.Z.-A.); ja.cinatl@kinderkrebsstiftung-frankfurt.de (J.C.); 3Department of Radiotherapy and Oncology, Goethe-Universität, 60590 Frankfurt am Main, Germany; fabian.weipert@gmail.com (F.W.); Franz.Roedel@kgu.de (F.R.); 4Department of Neuropathology, Ruprecht-Karls-University Heidelberg and Deutsches Krebsforschungszentrum, 69120 Heidelberg, Germany; andreas.vondeimling@med.uni-heidelberg.de; 5Division Neuroblastoma Genomics, B087, German Cancer Research Center and Hopp Children’s Cancer Center at the NCT (KiTZ), 69120 Heidelberg, Germany; f.westermann@dkfz-heidelberg.de

**Keywords:** YM155, survivin, neuroblastoma, drug resistance, ABCB1, ABCC1

## Abstract

The survivin suppressant YM155 is a drug candidate for neuroblastoma. Here, we tested YM155 in 101 neuroblastoma cell lines (19 parental cell lines, 82 drug-adapted sublines). Seventy seven (77) cell lines displayed YM155 IC_50_s in the range of clinical YM155 concentrations. ABCB1 was an important determinant of YM155 resistance. The activity of the ABCB1 inhibitor zosuquidar ranged from being similar to that of the structurally different ABCB1 inhibitor verapamil to being 65-fold higher. ABCB1 sequence variations may be responsible for this, suggesting that the design of variant-specific ABCB1 inhibitors may be possible. Further, we showed that ABCC1 confers YM155 resistance. Previously, p53 depletion had resulted in decreased YM155 sensitivity. However, *TP53*-mutant cells were not generally less sensitive to YM155 than *TP53* wild-type cells in this study. Finally, YM155 cross-resistance profiles differed between cells adapted to drugs as similar as cisplatin and carboplatin. In conclusion, the large cell line panel was necessary to reveal an unanticipated complexity of the YM155 response in neuroblastoma cell lines with acquired drug resistance. Novel findings include that ABCC1 mediates YM155 resistance and that YM155 cross-resistance profiles differ between cell lines adapted to drugs as similar as cisplatin and carboplatin.

## 1. Introduction

The inhibitor of apoptosis protein (IAP) survivin has multifaceted roles in cellular signalling. It is absent from most somatic cells but highly abundant in cancer cells and mediates cancer cell survival and proliferation. Elevated survivin levels have been associated with more aggressive and advanced cancer disease and lower survival rates. Hence, survivin is a potential drug target in cancer entities including neuroblastoma [1,2,3,4,5,6,7,8,9,10], the most frequent solid extracranial paediatric cancer. About half of the patients are diagnosed with high-risk disease associated with overall survival rates below 50% despite myeloablative therapy and differentiation therapy using retinoids. While many neuroblastomas respond initially well to therapy, acquired drug resistance represents a major clinical problem [11,12].

YM155 (sepantronium bromide) was introduced as a suppressor of survivin expression that displayed anti-cancer activity in pre-clinical models of different cancer entities including neuroblastoma [6,7,8,10,13]. However, DNA damage induction and Mcl-1 depletion were later suggested as additional or alternative anti-cancer mechanisms of YM155 [8,14,15,16,17,18]. We recently confirmed that YM155 exerts its anti-neuroblastoma effects predominantly through survivin suppression [10]. YM155-induced survivin suppression proceeded DNA damage formation. Moreover, YM155 mimicked the effects of RNAi-mediated survivin depletion, whereas Mcl-1 depletion did not affect neuroblastoma cell viability [10]. Furthermore, YM155-adapted UKF-NB-3 neuroblastoma cells had developed resistance to RNAi-mediated survivin depletion [10]. We also investigated the effects of YM155 in neuroblastoma cells with acquired resistance to cisplatin, doxorubicin, or vincristine and detected in concert with previous findings reduced SLC35F2 (mediates cellular YM155 uptake) and increased ABCB1 (causes YM155 efflux) expression as drug-specific resistance mechanisms [6,10,19,20]. Our data further demonstrated that RNAi-mediated p53 depletion mediated resistance to YM155 and survivin depletion suggesting loss of p53 function to be a target-specific resistance mechanism that will affect all approaches that target survivin in neuroblastoma [10].

Here, we investigated the effects of YM155 in a larger panel of 101 neuroblastoma cell lines that focused on acquired drug resistance containing 82 drug-adapted neuroblastoma cell lines, which reflected resistance to 15 anti-cancer drugs.

## 2. Results

### 2.1. Effects of YM155 on the Viability of Parental Neuroblastoma Cell Lines

YM155 was initially tested in a panel of 86 neuroblastoma cell lines consisting of 17 parental neuroblastoma cell lines and 69 drug-adapted sub-lines (Appendix A). The IC_50_ values in the parental cell lines ranged from 0.49 nM (UKF-NB-3) to 248 nM (LAN-6) (Figure 1, Appendix A). All parental cell lines despite from LAN-6, NB-S-124 (77 nM), and SK-N-SH (75 nM) had IC_50_ values in the range of clinical achievable YM155 plasma concentrations (Figure 1A, Appendix A) that were reported to reach up to 56 nM [21,22,23].

### 2.2. ABCB1-Expressing Neuroblastoma Cell Lines Display Low Sensitivity to YM155

In concert with previous studies [6,10], high ABCB1-expressing neuroblastoma cells generally displayed relatively low YM155 sensitivity (Figure 1A). In a subset of these cell lines, only the high ABCB1-expressing cells were (in contrast to low ABCB1-expressing cells) sensitised to YM155 by verapamil and zosuquidar (Figure 1B), two structurally unrelated ABCB1 inhibitors [24].

### 2.3. MYCN Status Does Not Influence Neuroblastoma Cell Sensitivity to YM155

MYCN amplification is a major determinant of poor disease outcome in neuroblastoma [11,12]. The YM155 IC_50_ ranged from 0.49 nM (UKF-NB-3) to 77 nM (NB-S-124) in MYCN-amplified cells and from 3.55 nM (SK-N-AS) to 248 nM (LAN-6) in not MYCN-amplified cells (Appendix A). To exclude the effects of ABCB1, we compared the YM155 IC_50_ in cells with known MYCN status in the presence of the ABCB1 inhibitors verapamil and zosuquidar. In the presence of ABCB1 inhibitors, MYCN-amplified and non-MYCN-amplified displayed a similar range of YM155 IC_50_s (Figure 2A, Appendix A). SH-EP-*MYCN* (TET21N) cells express a tetracycline-controllable MYCN transgene. They display low MYCN levels in the presence of tetracycline antibiotics and high MYCN levels in the absence of tetracycline antibiotics [25]. SH-EP-*MYCN* (TET21N) cells displayed similar YM155 IC_50_ values in the absence or presence of doxycycline (Figure 2B, Appendix A).

### 2.4. TP53 Status Does Not Predict Neuroblastoma Cell Sensitivity to YM155

Previously, RNAi-mediated p53 depletion was shown to reduce the YM155 sensitivity of the neuroblastoma cell lines UKF-NB-3 and UKF-NB-6 [10]. However, the p53-null SK-N-AS cells displayed an YM155 IC_50_ of 3.55 nM that was further reduced to 1.01 nM and 1.31 nM by verapamil and zosuquidar, respectively (Figure 1, Appendix A). Hence, SK-N-AS belongs in the presence of ABCB1 inhibitors to the most YM155-sensitive neuroblastoma cell lines in the panel, despite the lack of functional p53. 

To further investigate the relevance of the *TP53* status for the neuroblastoma cell sensitivity to YM155, we determined YM155 IC_50_ values in a panel of 14 nutlin-3-adapted *TP53*-mutant neuroblastoma cell lines [26,27]. Our initial cell line panel included one nutlin-3-adapted neuroblastoma cell line (UKF-NB-3^r^Nutlin^10µM^) that harbours a G245C loss-of-function *TP53* mutation [26] and displayed 2.4-fold reduced YM155 sensitivity relative to the parental UKF-NB-3 cells (Appendix A). In addition, we tested YM155 in nutlin-3-resistant, *TP53* mutant sub-lines of two clonal p53 wild-type UKF-NB-3 sub-lines (UKF-NB-3clone1, UKF-NB-3clone3) and the *TP53* wild-type neuroblastoma cell line UKF-NB-6 (Figure 3, Appendix A). Only four out of the 14 nutlin-3-resistant neuroblastoma cell lines displayed a > 2-fold change in the YM155 IC_50_ relative to the respective parental cells, with 3.3 (UKF-NB-3clone1^r^Nutlin^10µM^I) being the highest fold change (Figure 3, Appendix A). These findings do not suggest the cellular *TP53* status to be a good predictor of neuroblastoma cell sensitivity to YM155.

### 2.5. Effects of YM155 on the Viability of Neuroblastoma Cell Lines with Acquired Drug Resistance

In a panel of 69 sub-lines of the neuroblastoma cell lines IMR-5, IMR-32, NGP, NLF, SHEP, UKF-NB-2, UKF-NB-3, and UKF-NB-6 with acquired resistance to drug classes including platinum drugs, vinca alkaloids, taxanes, alkylating agents, topoisomerase I inhibitors, topoisomerase II inhibitors, and nucleoside analogues (Appendix A), resistance was commonly associated with decreased YM155 sensitivity. However, 48 resistant cell lines displayed YM155 IC_50_ values in the range of therapeutic plasma levels (up to 56 nM) (Appendix A).

Forty one (41) of the resistant cell lines (60%) displayed cross-resistance to YM155 (YM155 IC_50_ resistant sub-line/ YM155 IC_50_ respective parental cell line > 2). Twelve of these cell lines showed a fold change YM155 IC_50_ resistant sub-line/ YM155 IC_50_ respective parental cell line of > 2 and < 10, 18 (26%) cell lines a fold change > 10 and < 100, and 11 (16%) cell lines a fold change >100. 20 (29%) resistant cell lines were similarly sensitive to YM155 like the respective parental cell lines (fold change < 2 and > 0.5). Seven (10%) resistant cell lines were more sensitive to YM155 than the respective parental cell lines (fold change < 0.5) (Appendix A). There were cell line-specific differences. For example, eight out of nine (89%) UKF-NB-3 sub-lines and nine out of 10 (90%) UKF-NB-6 sub-lines, but only two out of 11 NLF sub-lines (18%) displayed cross-resistance to YM155 (Appendix A).

The YM155 IC_50_ values in the drug-resistant cell lines ranged from 0.40 nM (UKF-NB-3^r^GEMCI^10^) to 21,549 nM (IMR-5^r^DOCE^20^) (Appendix A). Drug class-specific differences in the YM155 resistance profiles can be observed, but the variation of the results is very large (Figure 4, Appendix A).

The groups differed in the fraction of cell lines that displayed YM155 IC_50_ values <56 nM. All of the eight parental, four alkylating agent-resistant, five topoisomerase I inhibitor-resistant, and six nucleoside analogue (gemcitabine)-resistant cell lines displayed YM155 IC_50_ values < 56 nM. Only 15 out of 20 (75%) platinum drug-adapted, nine out of 14 (64%) topoisomerase II inhibitor-adapted, five out of 12 (42%) vinca alkaloid-adapted, and two out of six (33%) taxane (docetaxel)-adapted cell lines exhibited YM155 IC_50_ values < 56 nM (Figure 4, Appendix A).

For the topoisomerase II inhibitor (doxorubicin, etoposide)- and platinum drug (carboplatin, cisplatin, oxaliplatin)-adapted cell lines, we had sufficient data to perform drug-specific analyses. For six cell lines, we had doxorubicin- and etoposide-resistant sub-lines (Figure 5A,B, Appendix A). To determine the mean YM155 IC_50_ value, we again removed the UKF-NB-3 sub-lines because of the high value of UKF-NB-3^r^DOX^20^. Results revealed that acquired doxorubicin resistance resulted in a generally more pronounced YM155 resistance phenotype than acquired etoposide resistance (Figure 5A,B, Appendix A).

In addition, the project cell line panel included carboplatin-, cisplatin-, and oxaliplatin-resistant sub-lines of five neuroblastoma cell lines (Figure 5C,D, Appendix A). Cisplatin and oxaliplatin resistance were associated with a lower degree of YM155 resistance than carboplatin resistance (Figure 5C,D, Appendix A).

### 2.6. Role of ABCB1 in the YM155 response of Drug-Adapted Neuroblastoma Cells

A significant amount of drug-adapted neuroblastoma cell lines displays increased ABCB1 activity [28], and ABCB1 has been previously shown to mediate YM155 resistance [6,10,19]. In agreement with previous results, transduction of neuroblastoma cells with ABCB1 resulted in YM155 resistance, which was reduced by siRNA-mediated ABCB1 depletion (Figure 6, Appendix A). Further, YM155 100 nM, a concentration that did not induce survivin depletion in ABCB1-transduced UKF-NB-3 (UKF-NB-3^ABCB1^) cells after 24h of incubation, reduced cellular survivin levels in UKF-NB-3^ABCB1^ cells in the presence of the ABCB1 inhibitors verapamil and zosuquidar (Figure 6, Appendix A). 

Hence, we further examined the effects of verapamil and zosuquidar on YM155 sensitivity in a panel of 60 drug-adapted neuroblastoma cell lines (Appendix A). Fifteen of the drug-adapted cell lines displayed an IC_50_ higher than 56 nM. In the presence of verapamil, only six of the drug-adapted cell lines displayed an IC_50_ higher 56 nM. In the presence of zosuquidar, only five of the drug-adapted cell lines displayed an IC_50_ higher than 56 nM. The YM155 IC_50_ values of eight cell lines (IMR-5^r^DOX^20^, IMR-5^r^VCR^10^, IMR^r^VINB^20^, NGP^r^DOX^20^, NGP^r^ETO^400^, UKF-NB-3^r^CARBO^2000^, UKF-NB-3^r^CDDP^1000^, UKF-NB-3^r^VINOR^40^) were reduced to levels below 56 nM by verapamil and zosuquidar. Notably, the fold sensitisation by ABCB1 inhibitors was low (verapamil: 1.2-fold, zosuquidar: 1.5-fold) in NGP^r^ETO^400^, although the YM155 IC_50_s were reduced below 56 nM. This suggests that ABCB1 expression is not a dominant YM155 resistance mechanism in NGP^r^ETO^400^ cells (Appendix A).

IMR-5^r^DOCE^20^, NGP^r^VCR^20^, and UKF-NB-2^r^CARBO^2000^ were sensitised by verapamil and zosuquidar to YM155. However, the effects of zosuquidar were more pronounced resulting in YM155 IC_50_ values below 56 nM, whereas the YM155 IC_50_ values remained above 56 nM in the presence of verapamil (Appendix A). In NLF^r^DOX^40^ and NLF^r^VINB^10^ cells, zosuquidar (but not verapamil) increased the YM155 IC_50_ values by mechanisms that appear to be unrelated to ABCB1 (Appendix A).

We selected two parental cell line/ drug-adapted subline pairs (IMR-5/IMR-5^r^DOCE^20^, IMR-32/ IMR-32^r^DOX^20^) for additional confirmatory experiments. The YM155 IC_50_s in these four cell lines in the absence and presence of zosuquidar determined MTT (used in the screen, measures oxidative phosphorylation in the mitochondria) were very similar to those determined by CellTiterGlo (alternative viability assay that measures ATP production) (Appendix A). Both drug-adapted sublines displayed increased ABCB1 levels (Appendix A). Moreover, YM155 500 nM, a concentration that did not affect cellular survivin levels and PARP cleavage in IMR-5^r^DOCE^20^ cells after 24h incubation, caused survivin depletion and PARP cleavage in IMR-5^r^DOCE^20^ cells in the presence of the ABCB1 inhibitors verapamil and zosuquidar (Figure 7, Appendix A).

### 2.7. ABCC1 Mediates Resistance to YM155

NLF^r^VCR^10^ cells were sensitised by verapamil but not by zosuquidar to YM155 (Appendix A). NLF^r^VCR^10^ cells are characterised by ABCC1 (also known as MRP1) expression but do not express ABCB1 [29]. Since only verapamil but not zosuquidar inhibits ABCC1 [30,31], this suggests that ABCC1 also mediates resistance to YM155. In agreement, the ABCC1 inhibitor MK571 substantially reduced YM155 sensitivity in ABCC1-expressing NLF^r^VCR^10^ cells but not in the parental NLF cell line that does not express ABCC1 (Appendix A).

### 2.8. Cross Resistance to YM155 is Caused by Multiple Resistance Mechanisms in Drug-Adapted Neuroblastoma Cells

Although our data indicate that ABCB1 plays an important role in the cross-resistance of drug-adapted neuroblastoma cell lines to YM155, additional mechanisms are also involved. Of the 60 drug-adapted cell lines, 35 displayed cross-resistance to YM155 (fold change YM155 IC_50_ resistant sub-line/ YM155 IC_50_ respective parental cell line > 2). In 24 of these drug-adapted cell lines, the YM155 IC_50_ remained > 2-fold higher in the presence of verapamil compared to the YM155 IC_50_ of the respective parental cell line in the presence of verapamil (Appendix A). Similarly, in 19 drug-adapted cell lines, the YM155 IC_50_ remained > 2-fold higher in the presence of zosuquidar compared to the YM155 IC_50_ of the respective parental cell line in the presence of zosuquidar (Appendix A). This included cell lines that were not sensitised by verapamil and/ or zosuquidar to YM155 and those that were sensitised to YM155 by verapamil and/ or zosuquidar but not to the level of the parental cells (Appendix A).

### 2.9. Discrepancies in the Effects of the ABCB1 Inhibitors Verapamil and Zosuquidar on Neuroblastoma Cell Sensitivity to YM155

Both, verapamil and zosuquidar, sensitised 25 of the drug-adapted cell lines to YM155 by >2-fold. There was an overlap of 23 cell lines that were sensitised to YM155 by both ABCB1 inhibitors by > 2-fold (Appendix A). Among the exceptions was the ABCC1-expressing cell line NLF^r^VCR^10^ that has already been described above. NGP^r^GEMCI^20^ cells were sensitised to YM155 by zosuquidar, whereas verapamil slightly increased the YM155 IC_50_ value in this cell line. In addition, UKF-NB-3^r^TOPO^20^ cells were sensitised to YM155 by verapamil by > 2-fold (fold change 3.7) but not by zosuquidar (fold change 1.4). UKF-NB-6^r^CARBO^2000^ cells were sensitised by zosuquidar to YM155 by > 2-fold (fold change 2.3) but not by verapamil (fold change 1.3) (Appendix A). The reasons for these differences remain unclear.

Since we had detected differences between the effects of the ABCB1 inhibitors verapamil and zosuquidar on the YM155 IC_50_ values in a number of cell lines, we compared both drugs in a wider panel of 74 cell lines (Figure 8, Appendix A). In 45 cell lines, the YM155 IC_50_ values were in a similar range in the presence of verapamil and zosuquidar (YM155 IC_50_ in the presence of verapamil/ YM155 IC_50_ in the presence of zosuquidar > 0.5 and < 2.0). This included 33 cell lines that were neither sensitised to YM155 by verapamil nor by zosuquidar and 12 cell lines that were sensitised by verapamil and zosuquidar by more than 2-fold and in a similar fashion (Appendix A). 

Thirteen cell lines were more strongly sensitised to YM155 by verapamil than by zosuquidar (YM155 IC_50_ in the presence of verapamil/YM155 IC_50_ in the presence of zosuquidar < 0.5). Only one of these cell lines (UKF-NB-6^r^ETO^200^) was sensitised by both verapamil and zosuquidar to YM155 by more than 2-fold. The remaining cell lines were either only sensitised to YM155 by more than 2-fold by verapamil and not by zosuquidar, or zosuquidar increased the YM155 IC_50_ by mechanisms that do not appear to be associated with effects on ABCB1 (Appendix A). 

Sixteen cell lines were more strongly sensitised to YM155 by zosuquidar than by verapamil (YM155 IC_50_ in the presence of verapamil/ YM155 IC_50_ in the presence of zosuquidar > 2.0). NGP^r^GEMCI^20^ cells were only sensitised by zosuquidar to YM155 but not by verapamil. The other 15 of these cell lines were sensitised by more than 2-fold to YM155 by both compounds with zosuquidar exerting more pronounced effects. The relative differences between these two drugs on the YM155 sensitivity of these cell lines ranged from 2.2 (NLF^r^DOCE^20^, UKF-NB-3^r^CARBO^2000^) to 65.2 (UKF-NB-3^r^DOCE^10^) (Appendix A).

## 3. Discussion

YM155 has been suggested as therapeutic option for the treatment of neuroblastoma including therapy-refractory disease [6,7,10]. In a larger cell line panel, mainly consisting of neuroblastoma cell lines with acquired drug resistance, we here show that 77 out of 101 tested neuroblastoma cell lines displayed YM155 IC_50_ values in the range of clinically achievable plasma concentrations up to 56 nM [21,22,23]. 

Although MYCN amplification is a major indicator of poor prognosis in neuroblastoma [11,12], the efficacy of YM155 was independent of the MYCN status. As previously shown [6,10,19], ABCB1-expressing cells displayed low YM155 sensitivity and were sensitised by the ABCB1 inhibitors verapamil and zosuquidar to YM155. The role of ABCB1 expression in neuroblastoma is not clear. ABCB1 expression at diagnosis is commonly regarded not to be of prognostic relevance [32], although ABCB1 expression has been reported in a significant fraction of patients [6,33,34]. Notably, six (35%) out of the 17 parental neuroblastoma cell lines investigated in this study (Be(2)C, LAN-6, NB-S-124, SHEP, SK-N-AS, SK-N-SH) are characterised by ABCB1 activity, which is similar to a previous study, although there was some overlap between the cell line panels [6]. Limited information is available on ABCB1 expression as acquired drug resistance mechanism in neuroblastoma. Many drug-adapted neuroblastoma cell lines display enhanced ABCB1 activity [27,35], and drug-adapted cell lines have been shown to reflect clinically relevant resistance mechanisms [26,36,37,38,39,40,41,42,43,44,45]. Some clinical hints may also point towards a role of ABCB1 in acquired resistance in neuroblastoma [36,46]. Hence, ABCB1 may represent an acquired resistance mechanism in neuroblastoma.

In contrast to ABCB1, ABCC1 (also known as MRP1) is generally accepted as prognostic factor in neuroblastoma [23]. Interestingly, NLF^r^VCR^10^ cells (that express ABCC1 but not ABCB1 [29]) were sensitised to YM155 by verapamil (inhibits ABCB1 and ABCC1 [27]) but not by zosuquidar (inhibits only ABCB1 [31]). Moreover, the ABCC1 inhibitor MK571 sensitised NLF^r^VCR^10^ cells to YM155. This suggests that ABCC1 mediates YM155 resistance.

Verapamil and zosuquidar further differed in their effects on neuroblastoma cell sensitivity to YM155. Zosuquidar protected some ABCB1-negative cell lines from YM155-induced toxicity by unknown mechanisms. More strikingly, we identified cell line-specific differences in the relative potencies of verapamil and zosuquidar on YM155 activity. Twelve ABCB1-expressing cell lines were similarly sensitised to YM155 by verapamil and zosuquidar. Few cell lines were stronger sensitised by verapamil than by zosuquidar. These data are difficult to interpret with regard to ABCB1 because verapamil interacts with a broader range of transporters than the specific ABCB1 inhibitor zosuquidar [30,31,47]. However, 15 cell lines were sensitised by more than 2-fold to YM155 by verapamil and zosuquidar with zosuquidar exerting up to 65-fold (UKF-NB-3^r^DOCE^10^) more pronounced effects than verapamil. Hence, the relative effects of verapamil and zosuquidar on ABCB1-mediated YM155 transport differ in individual cell lines from similar efficacy (in 12 cell lines) to 65-fold increased zosuquidar efficacy over verapamil. Since zosuquidar is regarded as highly specific ABCB1 inhibitor [31,47], this difference seems to depend on discrepancies in the interaction with ABCB1. Notably, ABCB1 polymorphisms and mutations may substantially alter ABCB1 substrate specificity [48,49], which may explain the cell line-specific variation in the relative influence of zosuquidar and verapamil on ABCB1 function. Hence, it might be possible to design ABCB1 inhibitors that preferentially target specific ABCB1 variants. In this context, we have previously shown that certain ABCB1 inhibitors preferentially interfere with the ABCB1-mediated transport of certain ABCB1 substrates [50].

Although RNAi-mediated p53 depletion decreases neuroblastoma cell sensitivity to YM155 as previously demonstrated [10], only 4 out of 14 nutlin-3-adapted *TP53*-mutant neuroblastoma cell lines displayed a >2-fold increased YM155 IC_50_ relative to the respective parental cell line. All nutlin-3-resistant cell lines remained sensitive to low nanomolar YM155 concentrations with IC_50_ values ranging from 0.40 to 1.50 nM. This shows that the role of p53 depends on the individual cellular context and that the *TP53* status on its own does not indicate neuroblastoma cell sensitivity to YM155. These data are in accordance with initial findings that reported the activity of YM155 to be unrelated to the *TP53* status [13,51].

The adaptation of neuroblastoma cell lines to drugs from different classes was associated with varying YM155 sensitivity profiles. Neuroblastoma cell adaptation to topoisomerase I inhibitors, the nucleoside analogue gemcitabine, or alkylating agents was not associated with a pronounced YM155 resistance phenotype. In contrast, neuroblastoma cell lines adapted to the taxane docetaxel or vinca alkaloids exhibited distinct cross-resistance to YM155. Topoisomerase II inhibitors and platinum drugs displayed intermediate potential to induce YM155 resistance.

For the topoisomerase II inhibitor- and platinum drug-adapted neuroblastoma cell lines, we could perform drug-specific sub-analyses. Among the topoisomerase II inhibitors, doxorubicin-resistant cells were more likely to be YM155-resistant than etoposide-resistant cells. This result may not be too surprising. Although doxorubicin and etoposide both inhibit the religation of the topoisomerase II cleavage complexes, they are structurally different compounds that differ in their exact interaction with this target and may exert additional varying effects [52,53,54]. Perhaps more strikingly, we also found differences among the carboplatin-, cisplatin-, and oxaliplatin-resistant cell lines. Carboplatin-resistant cells were more frequently YM155-resistant than cisplatin- or oxaliplatin-resistant cells. This is surprising because platinum drugs share a similar mechanism of action. In particular, the effects of carboplatin and cisplatin are thought to be much more related to each other than to oxaliplatin [55,56]. The underlying reasons remain unclear, but the findings indicate substantial gaps in our understanding of the action of these frequently used drugs that need to be filled.

## 4. Materials and Methods 

### 4.1. Drugs

YM155 (sepantronium bromide) and zosuquidar were purchased from Selleck Chemicals via BIOZOL Diagnostica GmbH (Eching, Germany), and verapamil was from Sigma-Aldrich (Munich, Germany).

### 4.2. Cells

The MYCN-amplified neuroblastoma cell lines UKF-NB-2, UKF-NB-3, UKF- and UKF-NB-6 were established from stage 4 neuroblastoma patients [26,57,58]. UKF-NB-3clone1 and UKF-NB-3clone3 are p53 wild-type single cell-derived sub-lines of UKF-NB-3 [26]. Be(2)C, IMR-32, SH-SY5Y, SK-N-AS, and SK-N-SH were obtained from ATCC (Manassas, VA, USA), CHP-134, LAN-6, NGP, and NMB from DMSZ (Braunschweig, Germany), and GI-ME-N from ICLC (Genova, Italy). IMR-5, NLF, and SHEP were kindly provided by Dr Angelika Eggert (Universität Duisburg-Essen, Germany). The MYCN-amplified NB-S-124 cell line was established by Dr Frank Westermann (DKFZ, Heidelberg, Germany). A master cell bank was established at the beginning of the project, and experiments on cell lines were performed within 20 passages. 

Resistant neuroblastoma cell lines were established by continuous exposure to increasing drug concentrations as previously described [26] and derived from the Resistant Cancer Cell Line (RCCL) collection (www.kent.ac.uk/stms/cmp/RCCL/RCCLabout.html).

All cells were propagated in Iscove’s modified Dulbecco’s medium (IMDM) supplemented with 10% foetal calf serum (FCS), 100IU/ml penicillin and 100µg/ml streptomycin at 37 °C. Drug-adapted cell lines were continuously cultivated in the presence of the respective adaptation drugs but were released from the respective adaptation drugs before they were used for experiments. Cells were routinely tested for mycoplasma contamination and authenticated by short tandem repeat profiling.

p53-depleted and ABCB1-transduced neuroblastoma cells were established as described previously [26] using the Lentiviral Gene Ontology (LeGO) vector technology (www.lentigo-vectors.de). SH-EP-*MYCN* (TET21N) were cultured and induced as previously described [25].

### 4.3. Viability Assay

Cell viability was determined by the 3-(4,5-dimethylthiazol-2-yl)-2,5-diphenyltetrazolium bromide (MTT) dye reduction assay as described previously [26] or by CellTiterGlo assay (Promega, Walldorf, Germany) following the manufacturer’s instructions after 120 h incubation.

### 4.4. Western Blot

Cells were lysed using Triton-X-100 sample buffer, and proteins were separated by SDS-PAGE. Detection occurred by using specific antibodies against β-actin (Biovision through BioCat GmbH, Heidelberg Germany), ABCB1, poly (ADP-ribose) polymerase (PARP) (both from Cell Signaling via New England Biolabs, Frankfurt, Germany), MYCN (abcam, Cambridge, UK), and survivin (R&D Systems, Minneapolis, MN, USA). Protein bands were visualised by laser-induced fluorescence using infrared scanner for protein quantification (Odyssey, Li-Cor Biosciences, Lincoln, NE, USA).

### 4.5. RNA Interference (RNAi)

Transient depletion of ABCB1 was achieved using synthetic siRNA oligonucleotides (ON-TARGETplus SMARTpool) from Dharmacon (Lafayette, CO; USA). Non-targeting siRNA (ON-TARGETplus SMARTpool) was used as negative control. Cells were transfected by electroporation using the NEON Transfection System (Invitrogen, Darmstadt; Germany) according to the manufacturer protocol. Cells were grown to 60–80 % confluence, trypsinised, and 1.2 x 10^6^ cells were re-suspended in 200 µl resuspension buffer R including 2.5 µM siRNA. The electroporation was performed using two 20 millisecond pulses of 1400 V. Subsequently, the cells were transferred into cell culture plates or flasks, containing pre-warmed cell culture medium.

### 4.6. TP53 Sequencing

*TP53* gene sequencing on cDNAs was performed using the following four pairs of primers: TP53 Ex2-3-f GTGACACGCTTCCCTGGAT and TP53 Ex2-3-r TCATCTGGACCTGGGTCTTC; TP53 Ex4-5-f CCCTTCCCAGAAAACCTACC and TP53 Ex4-5-r CTCCGTCATGTGCTGTGACT; TP53 EX6-7f GTGCAGCTGTGGGTTGATT and TP53 Ex6-7r GGTGGTACAGTCAGAGCCAAC; Tp53 Ex8-9-f CCTCACCATCATCACACTGG and TP53 Ex8-9-r GTCTGGTCCTGAAGGGTGAA. In addition, all cell lines were examined for TP53 mutations by sequence analysis of genomic DNA as described previously [26]. PCR was performed as described before [26]. Each amplicon was sequenced bidirectionally.

### 4.7. Statistics

Results are expressed as mean ± S.D. of at least three experiments. Comparisons between two groups were performed using Student’s t-test. Three or more groups were compared by ANOVA followed by the Student-Newman-Keuls test. P values lower than 0.05 were considered to be significant. 

## 5. Conclusions

The investigation of YM155 in 101 neuroblastoma cell lines revealed complex sensitivity profiles and that larger panels of model systems are needed to uncover this complexity. Our findings confirm 1) that YM155 is a drug candidate for neuroblastoma including therapy-refractory disease with the majority of cell lines being sensitive to clinically achievable YM155 concentrations and 2) that ABCB1 is an important determinant of YM155 sensitivity. Notably, there were substantial differences in the relative efficacy of the ABCB1 inhibitors verapamil and zosuquidar in sensitising ABCB1-expressing cells to YM155. These differences may be caused by sequence variations in the transporters in the different cell lines, which suggests that it may be possible to design variant-specific ABCB1 inhibitors. Moreover, we present novel findings indicating 1) that YM155 resistance is also mediated by ABCC1 (an ABC transporter of prognostic relevance in neuroblastoma [32]), 2) that (in contrast to previous assumptions) the p53 status does not indicate YM155 sensitivity, and 3) that YM155 cross-resistance profiles differ between cell lines adapted to drugs from different classes and even between cell lines adapted to drugs as similar as cisplatin and carboplatin. YM155 has shown moderate effects in clinical trials so far [23,59,60]. An improved understanding of the complex processes underlying response to this drug may enable the identification of biomarkers and the design more effective personalised therapies.

## Figures and Tables

**Figure 1 cancers-12-00577-f001:**
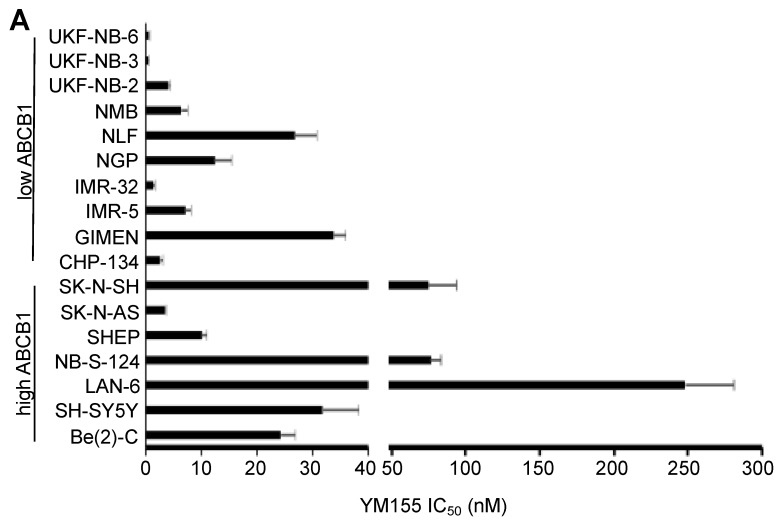
Anti-neuroblastoma effects of YM155 in a panel of 17 neuroblastoma cell lines. (**A**) YM155 concentrations that reduce the viability of neuroblastoma cell lines by 50% (IC_50_, mean ± S.D., n=3) as determined by MTT assay after a 5-day treatment period. Numerical values are presented in Appendix A. Information on the ABCB1 status of the cell lines is provided in Appendix A. (**B**) Effects of the ABCB1 inhibitors verapamil (5 µM) and zosuquidar (1.25 µM) on the YM155 IC_50_ values in neuroblastoma cell lines characterised by high or low ABCB1 levels displayed as fold change YM155 IC_50_/ YM155 IC_50_ in the presence of ABCB1 inhibitor. Numerical data and the effects of the ABCB1 inhibitors alone on cell viability are presented in Appendix A.

**Figure 2 cancers-12-00577-f002:**
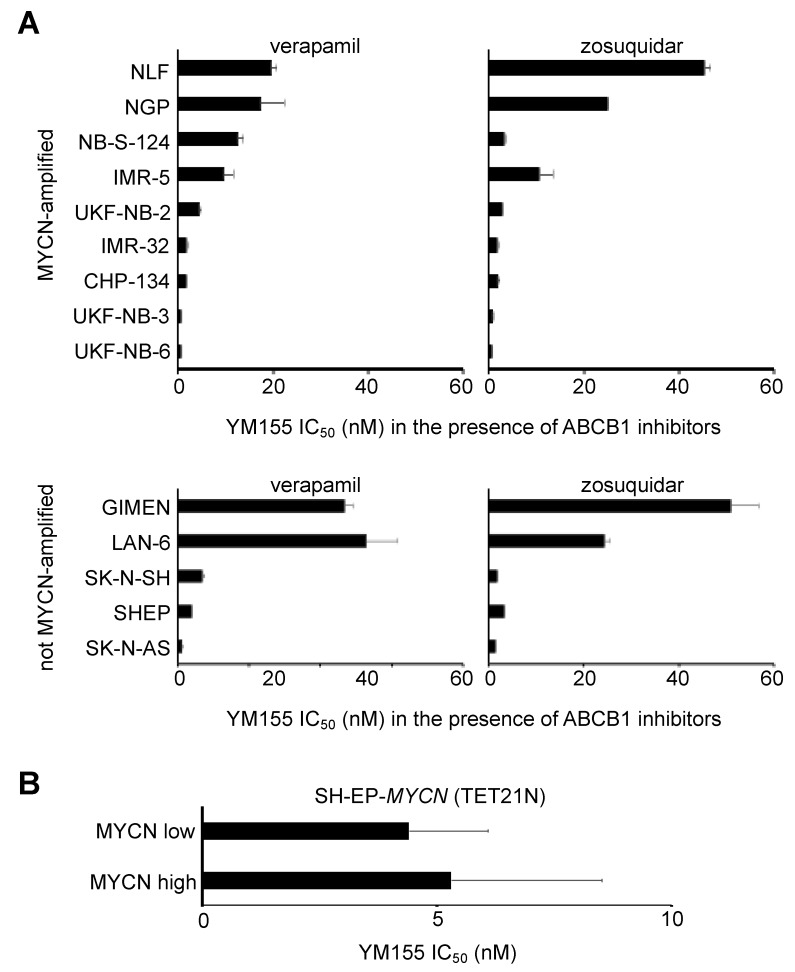
Effects of YM155 on the viability of neuroblastoma cells in dependence on the MYCN status. (**A**) YM155 concentrations that reduce the viability of neuroblastoma cell lines by 50% (IC_50_) were determined by MTT assay after a 5-day treatment period in the presence of the ABCB1 inhibitors verapamil (5 µM) or zosuquidar (1.25 µM) to avoid interference of ABCB1-mediated effects with MYCN-mediated effects. Numerical data are presented in Appendix A. (**B**) YM155 IC_50_ values in SH-EP-*MYCN* (TET21N) cells in the absence or presence of doxycycline as determined by MTT assay after a 120h of treatment. All values are presented as mean ± S.D. (n = 3).

**Figure 3 cancers-12-00577-f003:**
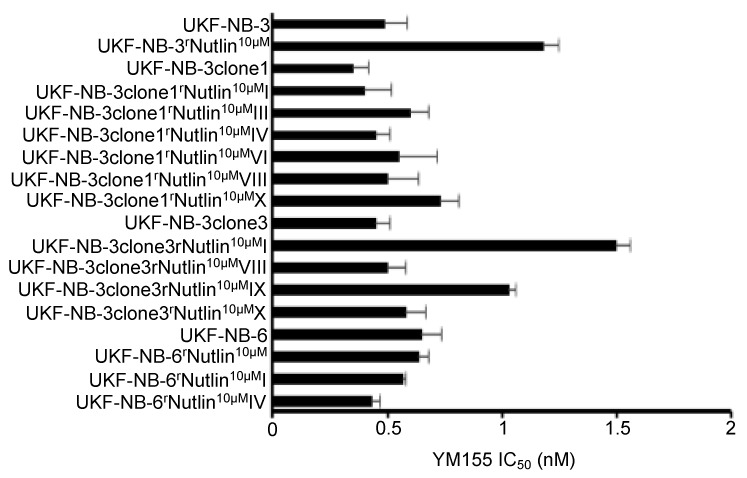
Effects of YM155 on the viability of parental p53 wild-type neuroblastoma cell lines and their p53 mutant nutlin-3-adapted sub-lines. YM155 concentrations that reduce neuroblastoma cell viability by 50% (IC_50_, mean ± S.D., n = 3) as determined by MTT assay after a 5-day treatment period. Numerical data are presented in Appendix A.

**Figure 4 cancers-12-00577-f004:**
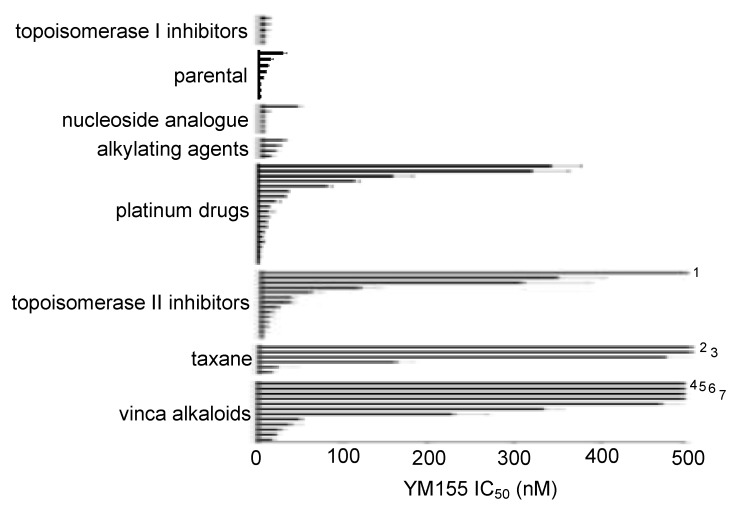
Effects of YM155 on the viability of neuroblastoma cell lines adapted to particular drug classes. Distribution of the YM155 IC_50_ values (mean ± S.D., n = 3) within the groups of drug-adapted cancer cell lines. Numerical data are presented in Appendix A. ^1^UKF-NB-3^r^DOX^20^ (YM155 IC_50_ 15,700 ± 1,019 nM); ^2^IMR-5^r^DOCE^20^ (YM155 IC_50_ 21,549 ± 638 nM); ^3^UKF-NB-2^r^DOX^20^ (YM155 IC_50_ 1,108 ± 179 nM); ^4^NGP^r^VCR^20^ (YM155 IC_50_ 6,986 ± 716 nM); ^5^UKF-NB-2^r^VCR^10^ (YM155 IC_50_ 5,940 ± 247 nM); ^6^IMR-5^r^VINOR^20^ (YM155 IC_50_ 4,978 ± 147 nM); ^7^IMR-5^r^VINB^20^ (YM155 IC_50_ 1,608 ± 212 nM).

**Figure 5 cancers-12-00577-f005:**
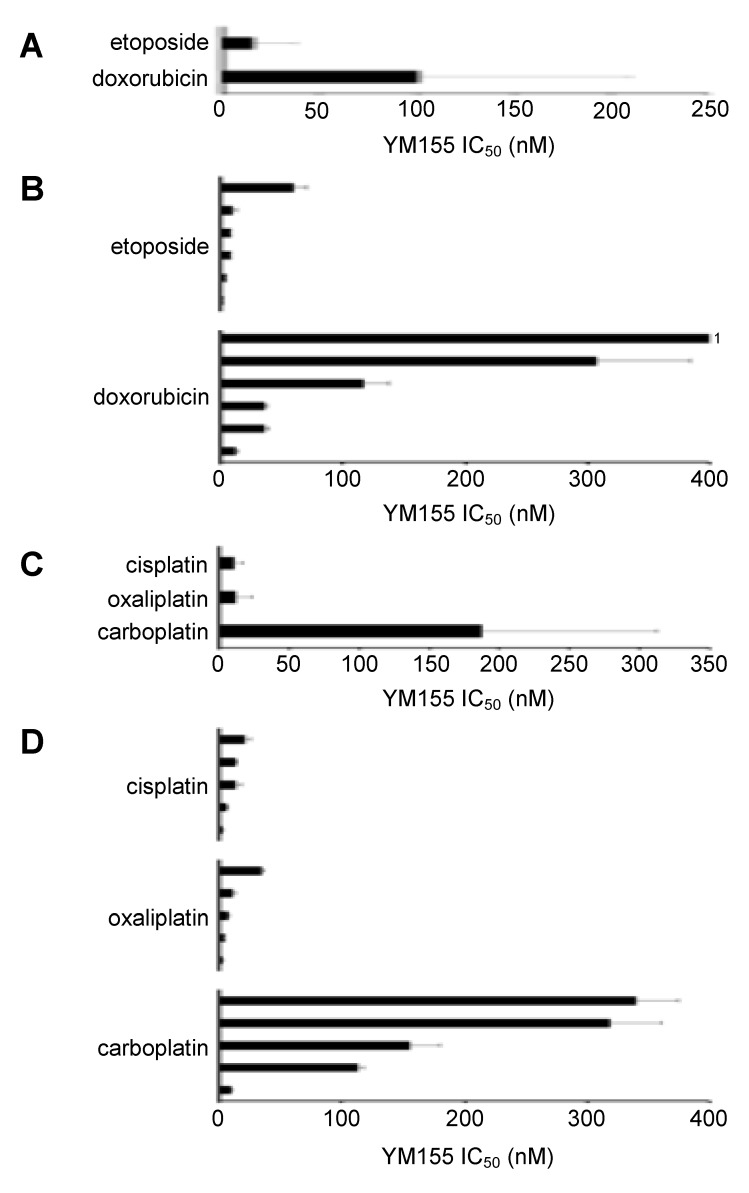
Effects of YM155 on the viability of neuroblastoma cell lines adapted to particular drugs. (**A**) YM155 concentrations that reduce the viability of neuroblastoma cell lines adapted to the topoisomerase II inhibitors doxorubicin or etoposide by 50% (IC_50_) as determined by MTT assay after a 5-day treatment period. Values are presented as mean ± S.D. over all cell lines from the individual groups. The cell line UKF-NB-3^r^DOX^20^ was not included into this analysis because it was regarded as outliers (please refer to the text). (**B**) Distribution of the YM155 IC_50_ values (mean ± S.D., n = 3) in doxorubicin- and etoposide-adapted cells. (**C**) YM155 IC_50_ concentrations in neuroblastoma cell lines adapted to the platinum drugs carboplatin, cisplatin, or oxaliplatin (mean ± S.D)) as determined by MTT assay after a 5-day treatment period. (**D**) Distribution of the YM155 IC_50_ values (mean ± S.D., n = 3) in carboplatin-, cisplatin- and oxaliplatin-adapted cells. Numerical data are presented in Appendix A. ^1^UKF-NB-3^r^DOX^20^ (YM155 IC_50_ 15,700 ± 1,019 nM).

**Figure 6 cancers-12-00577-f006:**
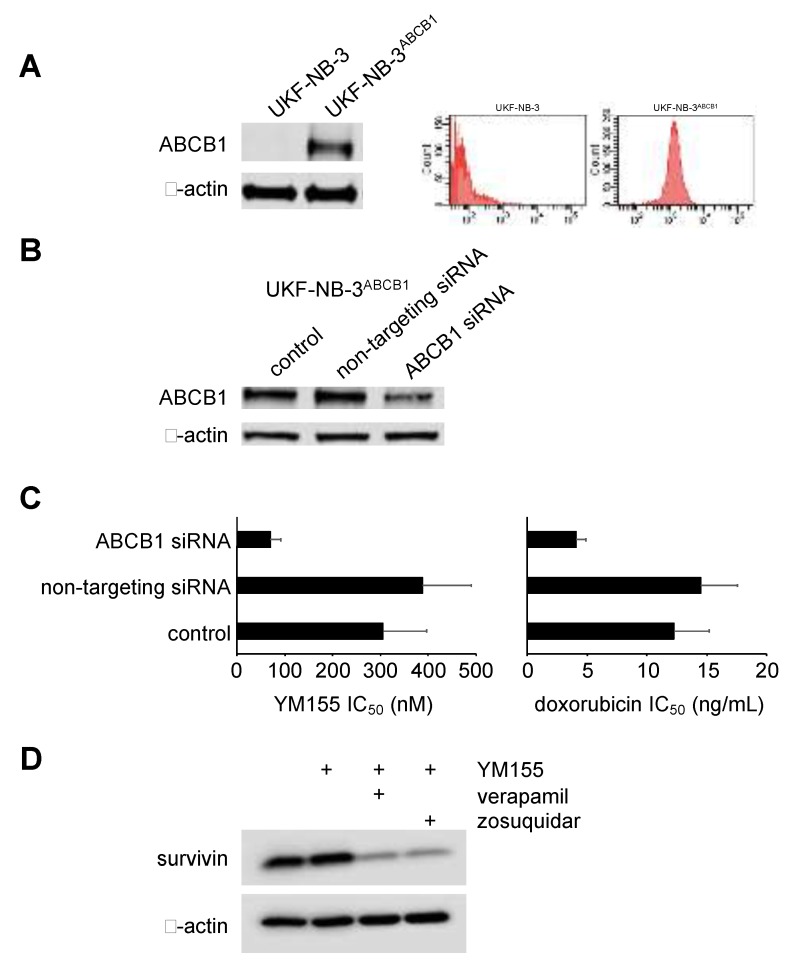
Effects of YM155 in ABCB1-transduced cells. (**A**) Representative western blots and flow cytometry histograms indicating ABCB1 levels in UKF-NB-3 cells and in UKF-NB-3 transduced with a lentiviral vector encoding ABCB1 (UKF-NB-3^ABCB1^). (**B**) Effect of siRNA directed against ABCB1 on cellular ABCB1 levels in UKF-NB-3^ABCB1^ cells. (**C**) Concentrations of YM155 and doxorubicin (alternative ABCB1 substrate used as control) that reduce the viability of UKF-NB-3^ABCB1^ cells by 50% (IC_50_, mean ± S.D., n = 3) as determined by MTT assay after 120h of incubation. (**D**) Effects of YM155 (100 nM) on survivin levels in UKF-NB-3ABCB1 cells after 24h of incubation in the presence or absence of verapamil (5 µM) or zosuquidar (1.25 µM). Uncropped Western blots are presented in Appendix A.

**Figure 7 cancers-12-00577-f007:**
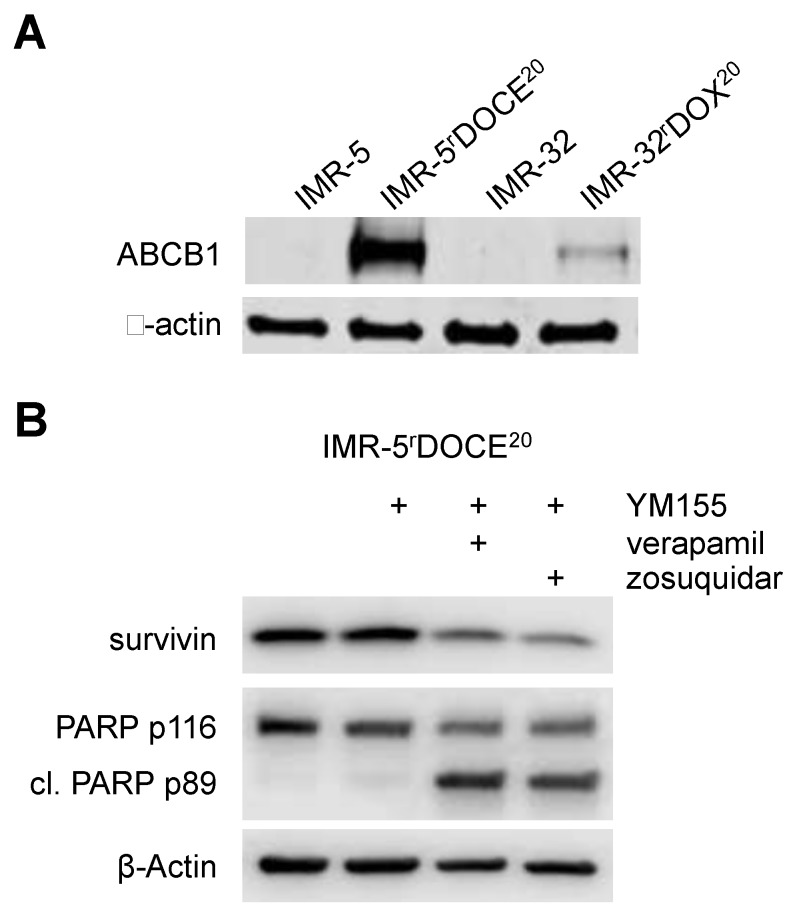
ABCB1 expression and YM155 activity in drug-adapted neuroblastoma cells. (**A**) Representative western blots indicating ABCB1 levels in IMR-5, IMR-5^r^DOCE^20^, IMR-32, and IMR-32^r^DOX^20^. (**B**) Effects of YM155 (500 nM) on survivin levels and PARP cleavage in IMR-5^r^DOCE^20^ cells in the presence or absence of verapamil (5 µM) or zosuquidar (1.25 µM) after 24h of incubation. Uncropped Western blots are presented in Appendix A.

**Figure 8 cancers-12-00577-f008:**
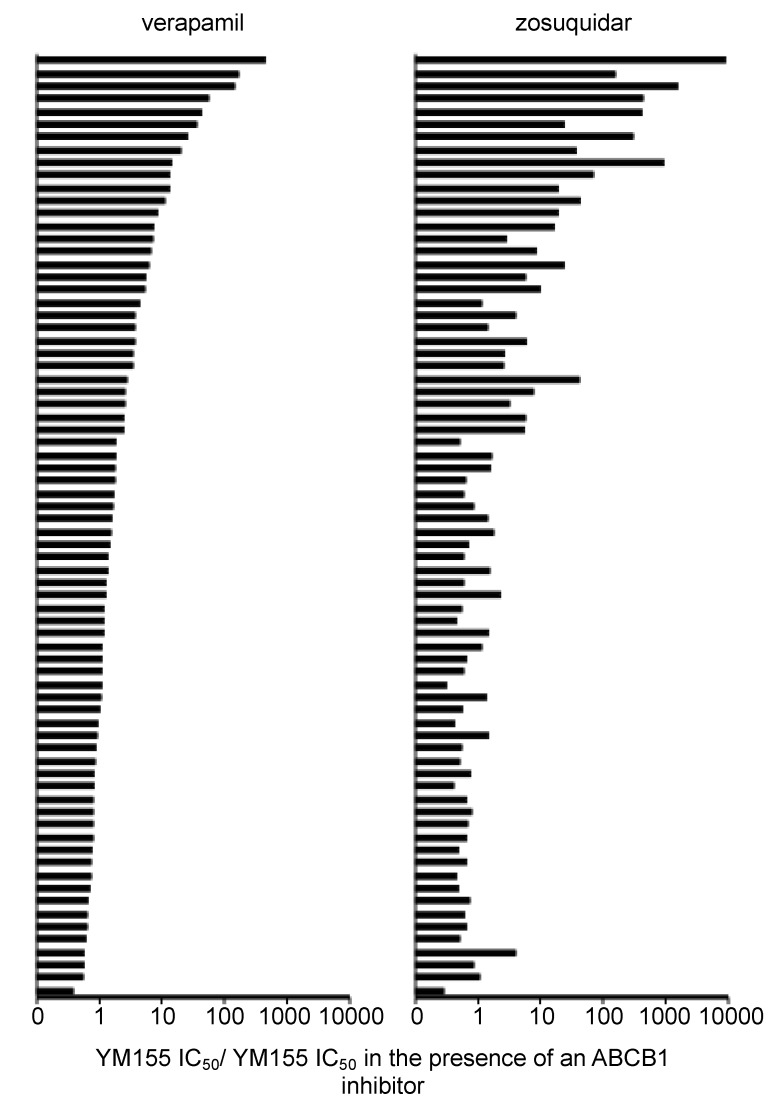
Comparison of verapamil- and zosuquidar-induced neuroblastoma cell sensitisation to YM155 in a panel of 74 neuroblastoma cell lines. The fold sensitisation (YM155 IC_50_/ YM155 IC_50_ in the presence of ABCB1 inhibitor) was determined by MTT assay after neuroblastoma cell incubation with YM155 for 120 h in the absence or presence of verapamil (5 µM) or zosuquidar (1.25 µM). Numerical data are presented in Appendix A. ^1^UKF-NB-3^r^DOX^20^, fold change 9235; ^2^IMR-5^r^DOCE^20^, fold change 1581; ^3^ UKF-NB-3^r^DOCE^10^, fold change 939.

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
