# Peer review of "Testing of the Survivin Suppressant YM155 in a Large Panel of Drug-Resistant Neuroblastoma Cell Lines"

_cancers, 2020, doi:10.3390/cancers12030577_

Round 1

Reviewer 1 Report

The manuscript by Michaelis et al. builds on their previous research published in Cell Death & Disease, 2016. The current study provides a comprehensive analysis of the impact of survivin suppressor, YM155, on viability of neuroblastoma cells in vitro. The major advantage of this study is the utilization of a large panel of cell lines, especially the chemotherapy-adapted cell lines, which allowed refining previously published results on YM155 effects in neuroblastoma. Although, in general, I find the study sound and worthy of publication, there are several issues (see below), which should be addressed in the revised version of the manuscript.

MAJOR ISSUES:

  1. Figure 1B: Western blot probing ABCB1 in all cell lines examined needs to be provided as panel C or as supplementary material to support the differences in ABCB1 expression and categorization into ABCB1-high and -low cell lines. Otherwise the study is based on the assumption that the utilized cell lines stably expressed same levels of ABCB1 as reported previously. Unfortunately, due to the in vitro culture conditions etc., this is often not the case and the expression levels need to be verified.
  2. Line 219: „2.7. YM155 is an ABCC1 substrate“. There is no evidence that would support this conclusion. The data presented suggest that ABCC1 is involved in the resistance to YM155; thus, I would agree with the conclusion that “ABCC1 mediates resistance to YM155”. However, as ABCC1 has a broad spectrum of substrates (it is not only a xenobiotic pump), this resistance might be caused indirectly by efflux of other molecules involved in the observed effects. To prove that YM155 is a substrate of ABCC1, the intracellular levels of YM155 in ABCC1-inhibited cells would have to be measured and compared with ABCC1 active control.
  3. A description of statistics, especially statistical tests used, and the number of technical and biological replicates for individual methods needs to be included in the Materials and Methods section. For each figure legend, please indicate what values are presented in the plots (? mean ± SD ?). Furthermore, I would suggest indicating the number of biological replicates (n=) in each figure legend, including supplementary figures.
  4. Figure 2 & Table S2: For each cell line, MYCN status needs to be at least supported by proper references.
  5. In the introduction, the molecular function of survivin should be briefly described to provide readers with a rationale why „survivin is a potential drug target in cancer“.

MINOR ISSUES

  1. Table S4: Please provide the legend for the numbers in brackets in YM155 IC50 columns (most probably indicated by “2” in the upper index). Furthermore, I found the table very chaotic. I would suggest 1) highlight the parental cell lines, 2) show two decimal places for all numbers and align the numbers to the right to help readers to appreciate the number value (more/less than parental cells?), 3) increase the table width to always fit YM155 IC50 values in one line.
  2. 1B: error bars should be included
  3. Line 87, “non-MYC-amplified”: Is this just a typo and it should read “non-MYCN-amplified”?
  4. Table S4 legend: individual values are presented in Table S1 not “Table 1”.
  5. 4A: Why authors do not show error bars? As clear from Table S4, standard deviation of the mean YM155 IC50 values for most drug classes is greater than the mean itself. I would suggest omitting the Figure 4A and Table S4 completely as they rather distract than provide meaningful information. Figure 4B provides much better overall image of the induction of cross-resistance.
  6. Figure 4: The quality of the image is poor. Error bars for Fig. 4B are hardly noticeable.
  7. Table S5: What are the values in brackets in IC50 column? The legend for upper indexes (1,2) used within the table is missing.
  8. All Tables should indicate what values are presented as IC50. Is it mean ± SD (as I suppose) or mean ± SEM or something else?
  9. Figure 6: Poor quality image, “zosuquidar” remained highlighted by spelling checker, most cell lines show extremely low change or no change in sensitization to YM155. Is it necessary to show all these values in the bar graph? If so, would not be a log scale better to plot the sensitization data? Otherwise, most of the bars are so tiny that it is nearly impossible to differentiate between 1-fold (no change) from 2-fold or 10-fold increase.
  10. 2 Cells: Please describe how was the resistance in the resistant neuroblastoma cell lines maintained and checked during the experiments. How were these cell lines processed prior to seeding for viability assays? In our hands, adaptation to drugs does not always lead to stable resistance and the cell lines revert to the non-resistant phenotype when the drug selective pressure is released.

Author Response

The manuscript by Michaelis et al. builds on their previous research published in Cell Death & Disease, 2016. The current study provides a comprehensive analysis of the impact of survivin suppressor, YM155, on viability of neuroblastoma cells in vitro. The major advantage of this study is the utilization of a large panel of cell lines, especially the chemotherapy-adapted cell lines, which allowed refining previously published results on YM155 effects in neuroblastoma. Although, in general, I find the study sound and worthy of publication, there are several issues (see below), which should be addressed in the revised version of the manuscript.

MAJOR ISSUES:

  1. Figure 1B: Western blot probing ABCB1 in all cell lines examined needs to be provided as panel C or as supplementary material to support the differences in ABCB1 expression and categorization into ABCB1-high and -low cell lines. Otherwise the study is based on the assumption that the utilized cell lines stably expressed same levels of ABCB1 as reported previously. Unfortunately, due to the in vitro culture conditions etc., this is often not the case and the expression levels need to be verified.

Authors’ response:

We do not feel that the protein levels would increase the significance of our study on this occasion. The ABCB1 status of these cell lines has been well established by others and us (Mol Cell Biol. 1989 Oct;9(10):4337-44; Ann Oncol. 1998 Sep;9(9):1009-14; Int J Cancer. 2003 Mar 10;104(1):36-43; Int J Oncol. 2005 Oct;27(4):1029-37; Int J Oncol. 2006 Feb;28(2):439-46; Cancer Lett. 2007 May 18;250(1):107-16; Eur J Cancer. 2012 Mar;48(5):763-71, Br J Cancer. 2014 Aug 12;111(4):716-25; PLoS One. 2014 Sep 30;9(9):e108758; Oncotarget. 2016 Sep 6;7(36):58051-58064; Cell Death Dis. 2016 Oct 13;7(10):e2410; Eur J Med Chem. 2016 Oct 21;122:744-755; J Assoc Med Sci 2018;51(2):72-80). In Figure 1B, only the ABCB1-expressing cells are sensitised by the ABCB1 inhibitors verapamil and zosuquidar to YM155. Since ABCB1 is well known to be an ABCB1 substrate (Eur J Cancer. 2012 Mar;48(5):763-71, Cell Death Dis. 2016 Oct 13;7(10):e2410; Sci Rep. 2017 Jun 8;7(1):3091), these findings provide functional confirmation of the known cellular ABCB1 phenotypes, which is why we are convinced that our data are valid.

  1. Line 219: „2.7. YM155 is an ABCC1 substrate“. There is no evidence that would support this conclusion. The data presented suggest that ABCC1 is involved in the resistance to YM155; thus, I would agree with the conclusion that “ABCC1 mediates resistance to YM155”. However, as ABCC1 has a broad spectrum of substrates (it is not only a xenobiotic pump), this resistance might be caused indirectly by efflux of other molecules involved in the observed effects. To prove that YM155 is a substrate of ABCC1, the intracellular levels of YM155 in ABCC1-inhibited cells would have to be measured and compared with ABCC1 active control.

Authors’ response:

We agree. The paragraph title was changed as suggested to “2.7 ABCC1 mediates resistance to YM155” (line 257).

  1. A description of statistics, especially statistical tests used, and the number of technical and biological replicates for individual methods needs to be included in the Materials and Methods section. For each figure legend, please indicate what values are presented in the plots (? mean ± SD ?). Furthermore, I would suggest indicating the number of biological replicates (n=) in each figure legend, including supplementary figures.

Authors’ response:

The information was added to the Figure an Table legends, and the following paragraph was added to the methods section (p. 16, line 445):

4.7 Statistics

Results are expressed as mean ± S.D. of at least three experiments. Comparisons between two groups were performed using Student’s t-test. Three or more groups were compared by ANOVA followed by the Student-Newman-Keuls test. P values lower than 0.05 were considered to be significant.”

  1. Figure 2 & Table S2: For each cell line, MYCN status needs to be at least supported by proper references.

      Authors’ response:

      The references were added to Table S2.

  1. In the introduction, the molecular function of survivin should be briefly described to provide readers with a rationale why „survivin is a potential drug target in cancer“.

Authors' response:

The first paragraph was amended as follows to do this (p. 1, line 37):

"The inhibitor of apoptosis protein  (IAP) survivin has multifaceted roles in cellular signalling. It is absent from most somatic cells but highly abundant in cancer cells and mediates cancer cell survival and proliferation. Elevated survivin levels have been associated with more aggressive and advanced cancer disease and lower survival rates.  Hence, survivin is a potential drug target in cancer entities including neuroblastoma [1-10], the most frequent solid extracranial paediatric cancer."

MINOR ISSUES

  1. Table S4: Please provide the legend for the numbers in brackets in YM155 IC50 columns (most probably indicated by “2” in the upper index). Furthermore, I found the table very chaotic. I would suggest 1) highlight the parental cell lines, 2) show two decimal places for all numbers and align the numbers to the right to help readers to appreciate the number value (more/less than parental cells?), 3) increase the table width to always fit YM155 IC50 values in one line.

      Authors' response:

      Presumably, this comment refers to Table S5. As requested, we highlighted the parental cell        line in italics and modified the table. The brackets indicate the 'fold sensitisation (YM155 IC50/ YM155 IC50 in the presence of ABCB1 inhibitor)', which is explained as footnote '2' below the        Table.

  1. 1B: error bars should be included

      Authors' response:

      The fold sensitisation in Figure 1B was calculated using the means of three independent experiments. Hence, there is no standard deviation. The respective standard deviations for    the IC50 values are provided in Table S5.

  1. Line 87, “non-MYC-amplified”: Is this just a typo and it should read “non-MYCN-amplified”?

      Authors' response:

      Many thank. Yes, this is a typo, which has been corrected.

  1. Table S4 legend: individual values are presented in Table S1 not “Table 1”.

      Authors' response:

      This was corrected.

  1. 4A: Why authors do not show error bars? As clear from Table S4, standard deviation of the mean YM155 IC50 values for most drug classes is greater than the mean itself. I would suggest omitting the Figure 4A and Table S4 completely as they rather distract than provide meaningful information. Figure 4B provides much better overall image of the induction of cross-resistance.

      Authors' response:

      As suggested, we have removed Figure 4A. The respective paragraph in the Results section       now reads (p. 7, line 148):

      "The YM155 IC50 values in the drug-resistant cell lines ranged from 0.40nM (UKF-NB-    3rGEMCI10) to 21,549nM (IMR-5rDOCE20) (Table S1). Drug class-specific differences in the        YM155 resistance profiles can be observed, but the variation of the results is very large    (Figure 4, Table S4)."

  1. Figure 4: The quality of the image is poor. Error bars for Fig. 4B are hardly noticeable.

      Authors' response:

      It has been an issue to add some of the Figures to the manuscript document in an           appropriate quality. We have included high-quality power point versions of the Figures into the supplements zip file.

  1. Table S5: What are the values in brackets in IC50 column? The legend for upper indexes (1,2) used within the table is missing.

      Authors' response:

      The footnotes are explained below the Table:

      1 effect of verapamil (5µM) or zosuquidar (1.25µM) alone on cell viability presented as     percentage (mean ± S.D.) relative to untreated control

         2 fold sensitisation (YM155 IC50/ YM155 IC50 in the presence of ABCB1 inhibitor)

  1. All Tables should indicate what values are presented as IC50. Is it mean ± SD (as I suppose) or mean ± SEM or something else?

      Authors' response:

      It is mean ± S.D. This was corrected.

  1. Figure 6: Poor quality image, “zosuquidar” remained highlighted by spelling checker, most cell lines show extremely low change or no change in sensitization to YM155. Is it necessary to show all these values in the bar graph? If so, would not be a log scale better to plot the sensitization data? Otherwise, most of the bars are so tiny that it is nearly impossible to differentiate between 1-fold (no change) from 2-fold or 10-fold increase.

      Authors' response:

      Many thanks. We have plotted the x-axis on a logarithmic scale, and it looks indeed better. It       has been an issue to add some of the Figures to the manuscript document in an appropriate        quality. We have included high-quality power point versions of the Figures into the           supplements zip file.

  1. 2 Cells: Please describe how was the resistance in the resistant neuroblastoma cell lines maintained and checked during the experiments. How were these cell lines processed prior to seeding for viability assays? In our hands, adaptation to drugs does not always lead to stable resistance and the cell lines revert to the non-resistant phenotype when the drug selective pressure is released.

      Authors' response:

      The drug-adapted sublines were continuously cultivated in the presence of drug to maintain         the resistance phenotype. However, the adaptation drugs were not present in the viability           assays. The method section was amended to clarify this (p. 16, line 408):

      "Drug-adapted cell lines were continuously cultivated in the presence of the respective    adaptation drugs but were released from the respective adaptation drugs before they were   used for experiments."

Reviewer 2 Report

Authors present a well-written manuscript that describes testing YM155, small molecule inhibitor of Survivin, on a large panel of neuroblastoma cell lines.

Major comments:
- Study is mostly based on IC50 values as experimental methodology.
The studied compound YM155 is an inhibitor of Survivin, a known negative regulator of apoptosis that blocks caspase activation. Yet no data on apoptosis/caspases was presented (i.e. immunoblotting, flow cytometry). Please address this issue. Additional experiments are welcome.
- Immunoblotting and Flow cytometry (S1, S2, S3) figures need to be moved to the main body of the manuscript (not supplement), as the manuscript itself can not consist only of IC50 bar charts.

Other comments:
- in the Introduction section please say a few words about Survivin (what kind of protein, its role and function, i.e. negative regulator of apoptosis)
- Fig.1 Please try to arrange cell lines according to low/high ABCB1 in Fig.1A, otherwise it is difficult to compare Fig.1A (untreated) and Fig.1B (treated).
- Line 78. “ABCB1-expressing” means “high ABCB1-expressing” ? in contrast to “low ABCB1-expressing”
- Fig.4A. No error bars are shown.
- Fig.4B. Graphs are too small and low resolution. Not clear what they represent. Authors might consider other figure format for showing these data.
- Fig.5. Graphs are low resolution. Please improve.
- Fig 5B and 5D. Bars need to be assigned.
- Line 199. “IC50s” -> “IC50 values”
- Fig.6. Remove red underlining of “zosuquidar”.

Typos:
- in Keywords. “suvivinG”->”survivin”

Author Response

Authors present a well-written manuscript that describes testing YM155, small molecule inhibitor of Survivin, on a large panel of neuroblastoma cell lines.

Major comments:
- Study is mostly based on IC50 values as experimental methodology.
The studied compound YM155 is an inhibitor of Survivin, a known negative regulator of apoptosis that blocks caspase activation. Yet no data on apoptosis/caspases was presented (i.e. immunoblotting, flow cytometry). Please address this issue. Additional experiments are welcome.

Authors’ response:

Apoptosis induction is actually shown in Figure 7 (former Figure S3). Given the heterogeneity of the drug response that we observe among the cell lines, one or two additional cell lines would be of limited significance. Only the analysis of all 101 cell lines would provide a comprehensive picture. Due to the scale of such experiments, we do not feel that this could and should be reasonably addressed in a revision and is, hence, rather subject of follow-up studies.

- Immunoblotting and Flow cytometry (S1, S2, S3) figures need to be moved to the main body of the manuscript (not supplement), as the manuscript itself can not consist only of IC50 bar charts.

Authors’ response:

We have moved Figure S2 (now Figure 6) and Figure S3 (now Figure 7) into the main manuscript. However, we feel that Figure S1 should remain a supplement, because it is a control experiment that is not required to follow the narrative of the manuscript.

Other comments:
- in the Introduction section please say a few words about Survivin (what kind of protein, its role and function, i.e. negative regulator of apoptosis)

Authors’ response:

The first paragraph was amended as follows to do this (p. 1, line 37):

"The inhibitor of apoptosis protein  (IAP) survivin has multifaceted roles in cellular signalling. It is absent from most somatic cells but highly abundant in cancer cells and mediates cancer cell survival and proliferation. Elevated survivin levels have been associated with more aggressive and advanced cancer disease and lower survival rates.  Hence, survivin is a potential drug target in cancer entities including neuroblastoma [1-10], the most frequent solid extracranial paediatric cancer."

- Fig.1 Please try to arrange cell lines according to low/high ABCB1 in Fig.1A, otherwise it is difficult to compare Fig.1A (untreated) and Fig.1B (treated).

Authors’ response:

This was done.

- Line 78. “ABCB1-expressing” means “high ABCB1-expressing” ? in contrast to “low ABCB1-expressing”

Authors’ response:
This was corrected (now line 84).

- Fig.4A. No error bars are shown.

Authors’ response:
As requested by another reviewer, we have removed this Figure due to limited significance. The data are still available in Table S4 for anyone who is interested in this analysis.

- Fig.4B. Graphs are too small and low resolution. Not clear what they represent. Authors might consider other figure format for showing these data.

Authors’ response:

It has been an issue to add some of the Figures to the manuscript document in an appropriate quality. We have included high-quality power point versions of the Figures into the supplements zip file.

- Fig.5. Graphs are low resolution. Please improve.

Authors’ response:

It has been an issue to add some of the Figures to the manuscript document in an appropriate quality. We have included high-quality power point versions of the Figures into the supplements zip file.

- Fig 5B and 5D. Bars need to be assigned.

Authors’ response:

We feel that this would reduce the legibility of the Figure. The numerical values are provided in Table S1.

- Line 199. “IC50s” -> “IC50 values”

Authors’ response:

This was changed (now line 230).

- Fig.6. Remove red underlining of “zosuquidar”.

Authors’ response:

This was done (now Figure 8).

Typos:
- in Keywords. “suvivinG”->”survivin”

Authors’ response:

This has been corrected.

Round 2

Reviewer 1 Report

I was pleased to read the revised version, which addressed most of my concerns.

However, I still feel that the ABCB1 status of individual neuroblastoma cell lines needs to be at least supported by proper references. This is not textbook information and cannot be presented without experimental evidence or referencing previous studies.

Please include the references for ABCB1 status individually for each of the cell lines in Fig. 1A or provide this as a supplementary table. Please also explain the rationale, preferably in the results section, why some cell lines from Fig. 1A were omitted from the analysis of sensitization by ABCB1 inhibitors shown in Fig. 1B. Although I believe this is not the case, the authors conclude that “ABCB1-expressing cells are sensitised by the ABCB1 inhibitors” and one must think why not all cell lines were analyzed and whether they were omitted intentionally.

Author Response

We added an additional supplementary Table (now Table S2) to address this concern. For three cell lines (NGP, LAN-6, NB-S-124), we added in this Table additional data showing their sensitivity to the alternative ABCB1 substrate vincristine in the absence or presence of the ABCB1 inhibitors verapamil and zosuquidar. The results provided additional functional confirmation to those obtained with YM155 regarding the respective ABCB1 phenotypes.

We randomly selected a subset of the cell lines from Figure 1A for subsequent experiments in Figure 1B. The Results section was modified to clarify this (line 83):

"In concert with previous studies [6,10], high ABCB1-expressing neuroblastoma cells generally displayed relatively low YM155 sensitivity (Figure 1A). In a subset of these cell lines, only the high ABCB1-expressing cells were (in contrast to low ABCB1-expressing cells) sensitised to YM155 by verapamil and zosuquidar (Figure 1B), two structurally unrelated ABCB1 inhibitors [24]."

Reviewer 2 Report

Authors did a good job by adding additional experimental data, i.e. immunoblots (including uncropped ones).

Issues have been addressed.

In my opinion the manuscript can be accepted for publication.

Author Response

Many thanks.